# Early *TP53* Alterations Shape Gastric and Esophageal Cancer Development

**DOI:** 10.3390/cancers13235915

**Published:** 2021-11-24

**Authors:** Pranshu Sahgal, Brandon M. Huffman, Deepa T. Patil, Walid K. Chatila, Rona Yaeger, James M. Cleary, Nilay S. Sethi

**Affiliations:** 1Department of Medical Oncology, Dana-Farber Cancer Institute, Boston, MA 02215, USA; pranshu_sahgal@dfci.harvard.edu (P.S.); brandon_huffman@dfci.harvard.edu (B.M.H.); james_cleary@dfci.harvard.edu (J.M.C.); 2Cancer Program, Broad Institute of Massachusetts Institute of Technology (MIT) and Harvard University, Cambridge, MA 02142, USA; 3Department of Medicine, Brigham and Women’s Hospital, Harvard Medical School, Boston, MA 02215, USA; 4Department of Pathology, Brigham and Women’s Hospital, Harvard Medical School, Boston, MA 02215, USA; dtpatil@bwh.harvard.edu; 5Tri-Institutional Program in Computational Biology and Medicine, Weill Cornell Medical College, New York, NY 10021, USA; chatilaw@mskcc.org; 6Marie-Josée and Henry R. Kravis Center for Molecular Oncology, Memorial Sloan Kettering Cancer Center, New York, NY 10065, USA; 7Department of Medicine, Memorial Sloan Kettering Cancer Center, New York, NY 10065, USA; yaegerr@mskcc.org; 8Gastrointestinal Cancer Center, Dana-Farber Cancer Institute, Boston, MA 02215, USA

**Keywords:** *TP53*, gastric cancer, esophageal cancer, early genomic events

## Abstract

**Simple Summary:**

Recent evidence establishes that gastric and esophageal (GE) adenocarcinomas are similar cancers at cellular, genomic, and epigenomic levels. Human GE adenocarcinomas develop *TP53* mutations at early stages of malignant progression. This contrasts with other gastrointestinal adenocarcinomas, including sporadic colorectal and pancreatic adenocarcinomas, where *TP53* alterations occur late. Exposure of the esophagus and stomach to environmental risk factors contributes to the selection of early *TP53* mutations and subsequent chromosomal instability, which then lead to activation of mitogen and cell cycle pathways in GE adenocarcinomas by way of focal amplifications rather than mutations. While early *TP53* mutations enable GE adenocarcinoma development, they also expose therapeutic vulnerabilities that should be prime for targeted therapy directed against the DNA damage response.

**Abstract:**

Gastric and esophageal (GE) adenocarcinomas are the third and sixth most common causes of cancer-related mortality worldwide, accounting for greater than 1.25 million annual deaths. Despite the advancements in the multi-disciplinary treatment approaches, the prognosis for patients with GE adenocarcinomas remains poor, with a 5-year survival of 32% and 19%, respectively, mainly due to the late-stage diagnosis and aggressive nature of these cancers. Premalignant lesions characterized by atypical glandular proliferation, with neoplastic cells confined to the basement membrane, often precede malignant disease. We now appreciate that premalignant lesions also carry cancer-associated mutations, enabling disease progression in the right environmental context. A better understanding of the premalignant-to-malignant transition can help us diagnose, prevent, and treat GE adenocarcinoma. Here, we discuss the evidence suggesting that alterations in *TP53* occur early in GE adenocarcinoma evolution, are selected for under environmental stressors, are responsible for shaping the genomic mechanisms for pathway dysregulation in cancer progression, and lead to potential vulnerabilities that can be exploited by a specific class of targeted therapy.

## 1. Gastric and Esophageal Adenocarcinomas Are Similar Cancers

Cellular, molecular, genetic, and epigenomic analyses of adenocarcinomas of the stomach and esophagus demonstrate that these cancers are highly related. Gastric and esophageal adenocarcinomas appear to arise from a shared tissue of origin. A mouse model of Barrett’s esophagus (BE), widely considered the precursor lesion to esophageal adenocarcinoma, showed that Lgr5+ gastric cardia stem cells are potentially the cells of origin [1]. Furthermore, esophageal adenocarcinomas display chromatin profiles that mirror those found in gastric tissue (*H3K9me3*, *H3K4me1*, *H3K4me3*, *H3K9ac*, *H3K36me3*) rather than in normal esophageal tissue [2]. Beyond the common ancestry of these cancers, deeper evaluation of human gastric and esophageal adenocarcinomas demonstrates shared molecular features. DNA methylation, mRNA and miRNA expression, and somatic copy-number alterations (SCNA) of esophageal adenocarcinomas are almost identical to gastric adenocarcinoma, and quite distinct from esophageal squamous cell carcinoma [3,4,5,6]. While a comprehensive molecular analysis of gastrointestinal adenocarcinomas revealed five molecular subtypes that largely transcended anatomic boundaries [7], the similarity in molecular composition of GE adenocarcinomas is largely due to the high proportion of the chromosomal instability (CIN) subtype, which accounts for more than half of gastric adenocarcinomas and almost 90% of esophageal adenocarcinomas. By appreciating the cellular and molecular resemblance between gastric and esophageal adenocarcinomas, not only can we generate better models for the premalignant and malignant states, but we can also think about treating patients with these cancers in similar ways.

## 2. *TP53* Alteration Is an Early Event in GE Premalignancy

*TP53* is the most frequently mutated gene across all cancers, especially advanced metastatic disease [8]. Among GE adenocarcinomas, the frequency of *TP53* mutations is particularly high and enriched in the CIN subtype, which is characterized by increased aneuploidy within the tumor cells [7,9]. While the high rate of recurrent *TP53* mutations is a common finding amongst all gastrointestinal adenocarcinomas, the timing of its occurrence in the malignant progression has significant, and biologically important, variation. Unlike other gastrointestinal adenocarcinomas, including sporadic colorectal and pancreatic adenocarcinomas, human GE adenocarcinomas develop *TP53* mutations early in neoplasia, often found in premalignant lesions [10,11]. If not directly altered, other factors that regulate *TP53* are also dysregulated, such as *MDM2* and *WWOX*, the latter of which is a tumor suppressor found in a common chromosomal fragile site, and a regulator of the DNA damage response, whose protein expression is absent in up to 65% of gastric adenocarcinomas [12,13,14,15].

Clinically, a major challenge in the management of BE and gastric intestinal metaplasia, the respective premalignant lesions of GE cancers, is predicting when a precursor lesion will develop into adenocarcinoma. The frequency of progression from BE to esophageal adenocarcinoma is quite low [16]; the rates of gastric intestinal metaplasia progression to cancer are challenging to measure [17] but are thought to be infrequent. While *TP53* mutations are known to be early events, genomic analysis of clinical samples of BE and gastric intestinal metaplasia have demonstrated that the frequency of *TP53* mutations is actually low (~2%) [18]. Despite the low prevalence of *TP53* mutations in premalignant lesions, their presence seems to preferentially influence progression to a malignant state. Using a paired-sampling approach, in which BE lesions and esophageal adenocarcinoma from the same patient were subjected to whole exome sequencing, it was shown that premalignant and malignant lesions from an individual patient shared a specific *TP53* mutation more often than other tumor suppressors [19]. Another study that utilized whole genome sequencing of 23 paired human BE and esophageal adenocarcinoma samples showed that the degree of aneuploidy increases in the progression to cancer, suggesting that early inactivation of *TP53* might be a critical step in allowing the development of aneuploidy [20]. Furthermore, a separate study demonstrated that 46% of BE lesions that progress to high grade dysplasia or esophageal adenocarcinoma harbor *TP53* mutations, whereas BE lesions that did not progress, only had *TP53* mutations 5% of the time [21]. These data suggest that the presence of *TP53* mutations in premalignant GE lesions, even nondysplastic tissue, is a biomarker that predicts a high risk for malignant progression.

While colorectal and GE adenocarcinomas exhibit CIN, the characteristic pattern of aneuploidy observed in these cancers is notably different. GE adenocarcinomas display high intensity amplifications involving narrower genomic regions, whereas CRC tumors manifest broader, lower intensity amplifications [7]. For example, high-level focal amplifications of genes like *MET*, *FGFR2*, and *HER2* are commonly seen in GE adenocarcinomas, whereas colorectal adenocarcinomas more typically exhibit low-level amplification of multiple genes neighboring each other on a chromosome. It can, therefore, be inferred that *TP53* alterations may have occurred earlier in the evolution of GE cancers, enabling the development of high-level focal amplification in oncogenes that are critical to the development of the cancer. GE precursor lesions that develop high-level focal amplification of essential oncogenes have more proficient growth and are positively selected over precursor lesions that lack amplification of these oncogenes (Figure 1, Table 1). In other words, neoplastic cells with CIN that undergo stochastic disruption of critical genes within an amplicon die, whereas those that preserve oncogenes survive and thrive, propagating selection of malignant cells with narrower amplicons, broadcasting desired genes that support cancer progression. Later, we will discuss how known oncogenes in cancer-promoting molecular pathways are likely to be amplified in GE cancer rather than mutated in a gain-of-function fashion, which is typically the predominant mechanism of alteration for oncogenes in CRC (e.g., *KRAS*). Together, these data suggest that the more fragmented aneuploidy observed in GE cancers may be another indication that *TP53* alteration is an early event in disease evolution.

## 3. Context Matters: Environmental Conditions Contribute to Selection of Early *TP53* Alterations

Why are early *TP53* mutations selected for in upper, relative to lower, GI adenocarcinomas? Is it because gastric and esophageal cells are more susceptible to *TP53* mutations than colorectal cells, implicating the cell of origin as the culprit? Or rather, is it the environmental context of the upper, compared to the lower, GI tract that selects for *TP53* mutations? An examination of sporadic versus colitis-associated cancer (CAC) argues that there is a substantial contribution from environmental context. CAC arises in the setting of inflammatory bowel disease (IBD): the colon and other parts of the GI tract are subject to relapsing bouts of inflammation. As opposed to sporadic colorectal cancer (CRC), which is typically initiated by somatic alterations in WNT pathway tumor suppressor *APC*, followed by alterations in *KRAS* and *SMAD4*, CAC demonstrates a distinct pattern of genomic alterations notable for early *TP53* mutations [45,46,47], and a significantly lower frequency of *APC* mutations [48,49]. In fact, nondysplastic tissue from patients with IBD, arising from chronic inflammation of the gastrointestinal tract, often demonstrates *TP53* alterations [50]. A recent study analyzed molecular alterations in low-grade dysplastic lesions arising within and outside segments of colon affected by ulcerative colitis, as well as sporadic adenomas from non-IBD patients, and showed that, while all three cohorts harbored mutations in *APC* and *CTNNB1*, *TP53* mutations were only seen in lesions within areas of known colitis, albeit at low frequencies [51]. These data indicate that environmental context plays a critical role in the selection of genome alterations that lead to premalignant and eventually malignant disease.

Premalignant lesions that develop in the setting of IBD also display more CIN than sporadic adenomas [52]. Phylogenetic analysis of genomic changes in dysplasia and cancer from patients with colitis suggests that copy-number alterations begin to accrue in non-dysplastic bowel; the transition to low-grade/high grade dysplasia, however, often involves a punctuated increase in copy-number alterations [53]. Whole-genome sequencing of colonic crypts from patients with IBD and healthy controls revealed that the number of copy-number variants and retrotranspositions were associated with IBD duration, and that accumulation of structural variants in patients with IBD often exhibited an episodic nature, consistent with rapid accrual in the transition from mucosa to high-grade dysplastic lesions [54]. Furthermore, similar to GE adenocarcinomas in which a greater degree of focal CIN and gene amplifications drive cancer pathogenesis, CAC also displays more fragmented genomes, relative to sporadic CRC [55]. Chronic inflammation is a hallmark in the development of both GE adenocarcinoma and CAC, and strikingly, both cancers display early *TP53* mutations, followed by high-level focal amplification of essential oncogenes.

The esophagus and the stomach are exposed to dietary contents that either directly imbue carcinogenic properties or indirectly lead to byproducts with carcinogenic attributes, such as nitrosamines, as well as bile reflux, which collectively contribute to the development of premalignant lesions [56,57,58]. Apart from dietary carcinogens, *Helicobacter pylori* infection is also a well-established risk factor for gastric cancer development [59]. Many human studies have demonstrated a relationship between Helicobacter-associated gastric neoplasia and the development of *TP53* mutations [59,60,61,62,63,64,65]. We suspected that these exogenous exposures provide selective pressures for the emergence of mutant *TP53* clones. To test this hypothesis, we recently developed an integrative mouse model that combines disease-relevant exposures, such as dietary carcinogens, with tissue-specific *TP53* alterations to study the development of GE premalignancy [66]. In this mouse model, we found that inducible p53 inactivation in the stomach alone did not lead to premalignant lesions; by contrast, p53 inactivation combined with dietary carcinogens led to the expansion of p53 mutant clones and a greater burden of premalignant lesions. The mechanism underlying the cooperation of environmental risk factors and early genomic alterations in GE premalignancy deserves further investigation.

Multiple signaling pathways were upregulated in premalignant p53-deleted gastric lesions of the integrated mouse model that developed in the setting of dietary carcinogen exposure [66]. Among the dysregulated pathways, WNT signaling was notably activated, as indicated by gene expression profiling of derivative organoids, and immunohistochemical staining of downstream mediators in gastric premalignant lesions. These results may explain the relatively lower frequency of activating WNT pathway alterations in GE cancers, compared to sporadic CRC. Supporting this notion, a study involving human gastric cancer organoids demonstrated that co-alteration of *TP53* and *CDH1* promoted WNT signaling, independent of WNT agonist R-spondin [67]. The mechanism underlying WNT activation without frequent genomic alterations in GE adenocarcinomas requires further investigation. By contrast, WNT signaling appears to be downregulated in CAC; for example, a study looking at β-catenin staining demonstrated a greater proportion of tumors with low levels of nuclear beta-catenin in CAC compared to CRC [53]. Investigative human and mouse models that incorporate *TP53* mutations and disease-relevant inflammation in the lower gastrointestinal tract may provide insight into the WNT-independent drivers of CAC.

## 4. Early *TP53* Mutations Shape the Method of Genomic Alterations in Cancer-Promoting Pathways

As discussed above, early *TP53* mutations enable the development of a type of chromosomal instability that yields a more fragmented, aneuploid cancer genome with characteristic focal amplifications in GE and CAC, compared to sporadic CRC. GE tumors with focal CIN are associated with genome doubling and poor prognosis [7,68]. Of note, these tumors also select for a distinct genomic pattern of pathway dysregulation, manifesting with high-level focal amplifications of unmutated genes involved in MAPK signaling (*ERBB2*, *KRAS*, *VEGFA*) and cell cycle regulation (*CCNE1*, *CDK6*) [4,69]. Similarly, CAC often displays amplifications in *ERBB2*, *FGFR1/2*, and *MYC* [48]. In contrast, sporadic tumors arising from the colon and rectum more typically alter these same pathways by oncogenic mutations. For example, when altered, *KRAS* is almost exclusively mutated in sporadic CRC, whereas an unmutated version is more commonly amplified, rather than mutated, in GE adenocarcinomas [4,20].

Tumors overexpressing wild-type *KRAS* appear to be more resistant to MAPK inhibition, based on preclinical evidence. One mechanism indicates that KRAS-amplified tumors augment signaling from upstream receptors [70] and, therefore, are prone to feedback reactivation in the setting of downstream MAPK pathway blockade, effectively attenuating the activity of MAPK inhibitors that target MEK and ERK. Another explanation for the observation that *KRAS*-amplified GE adenocarcinomas are resistant to conventional MAPK pathway blockade is due to a potent adaptive response involving MEK signaling [71]. This adaptive response to MAPK inhibitors is diminished by combining pharmacological targeting of both SHP2 and MEK. While the mechanism for the increased efficacy of the SHP2 and MEK inhibitor combination is not clear, it is possible that SHP2 blocks RTK feedback loops, which lead to resistance in MEK inhibitor monotherapy [71]. These preclinical data provide rationale for utilization of MEK inhibitor/SHP2 inhibitor combinations in KRAS-amplified GE adenocarcinomas, which are currently being tested in multiple clinical trials. Overall, these data indicate that the early onset of CIN in GE adenocarcinoma and CAC leads to a distinct genomic pattern of cancer pathway alterations by selectively amplifying MAPK and cell cycle genes, as opposed to mutating them, which may carry implications for an alternative treatment approach.

## 5. Therapeutic Vulnerabilities Imparted by Early *TP53* Mutations

Among the molecular subtypes of GE adenocarcinomas, CIN is the largest and most heterogeneous, accounting for more than 50% of gastric cancers and almost all esophageal adenocarcinomas [3,9]. The shared genomic background of gastric and esophageal adenocarcinomas has led to the clinical recognition that these cancers should be considered similar, and hence, patients are often referred to as having “gastroesophageal adenocarcinoma”. Furthermore, the shared biology of gastric and esophageal adenocarcinoma has led to clinical trials being designed so that both populations are included, since the response rates to targeted and immunotherapies are often similar. CIN tumors harbor recurrent *TP53* mutations, display marked aneuploidy, and often select for amplification in receptor tyrosine kinase pathways (e.g., *EGFR* and *ERBB2*). While these methods of pathway activation promote cancer progression, CIN may also yield therapeutic vulnerabilities [72]. Genomic instability from CIN leads to the accumulation of genomic aberrations, which promotes rapid cell division and hampers regulatory mechanisms designed to control the cell cycle. These events can generate significant pressure on the DNA replication process, with resultant stalled or collapsed DNA replication forks, which is termed replicative stress. In response to replicative stress, compensatory proteins in the DNA damage response (DDR) pathway, such as ATR and CHK1, become activated to try to stabilize the DNA replication forks. Activation of these checkpoints in response to DNA damage and replication stress may, therefore, generate new sensitivities in specific genomic contexts. p53-deficient cancer cells have exhibited a selective sensitivity to inhibition of CHK1 when treated with cytotoxic agents or gamma-radiation [73]. Upstream of CHK1, ATR inhibition imparted selective toxicity in ATM- and p53-deficient cancer cells [74].

In addition to p53 deficiency, genomic alterations of other genes involved in cell cycle regulation may further increase sensitivity to inhibitors of cell cycle checkpoints. For example, amplification of *CCNE1* and *MYC*, as well as *FBXW7* mutations, are known to increase replicative stress. Hence, these alterations may be biomarkers of increased sensitivity to therapeutic strategies utilizing ATR, CHK1, and WEE1 inhibitors. Ongoing preclinical research is aimed at developing additional biomarkers that could predict sensitivity to ATR, CHK1, and WEE1 inhibitors [9,66,75]. Although ovarian cancer harbors *TP53* mutations in over 95% of cases, the CHK1/2 inhibitor prexasertib demonstrated a modest clinical response in a phase II clinical trial of BRCA wild-type ovarian cancers [76,77,78]. DDR inhibitors have yielded modest tumor responses in early phase clinical trials, except for a handful of cases that exhibit exquisite sensitivity, suggesting that predictive biomarkers will help identify patient populations that will benefit the most [79,80]. Other than the PARP inhibitor trials, there is a paucity of data available in GE adenocarcinoma regarding sensitivity to DDR inhibitors, despite their therapeutic potential; therefore, we need a deeper understanding of DDR pathway inhibitors in p53 mutant models of GE cancer.

The success of PARP inhibitors in targeting homologous recombination deficiency in other malignancies has generated interest in exploring whether PARP inhibitors could be effective in GE adenocarcinomas. While inactivating *BRCA1/2* and *PALB2* mutations are relatively rare, ATM loss occurs in up to 15–20% of gastric adenocarcinomas [81], although this may be an overestimation. In the GOLD randomized phase III clinical trial, treatment of advanced gastric cancer patients with the PARP inhibitor olaparib and paclitaxel did not meet its primary endpoint of improving survival in the overall study population nor in patients with ATM-deficient tumors in the second-line setting [82]. In this trial, ATM deficiency was defined immunohistochemically as observing ATM staining in less than 25% of tumor cell nuclei. While targeting ATM deficiency with PARP inhibition has, thus far, been unsuccessful, recently, a phase I trial of BAY-1895344 ATR inhibitor monotherapy showed encouraging efficacy in ATM-deficient tumors [83]. In this trial, an objective radiological response was observed in three solid tumor patients with ATM loss (defined as observing ATM staining in less than 1% of tumor cell nuclei) (Figure 2). Based on these results, better predictive and functional biomarkers of p53 mutant GE cancers that are sensitive to DDR pathway inhibitors are required for effective translation.

## 6. Conclusions

In this perspective article, we provided multiple lines of evidence that GE neoplasia harbor early *TP53* mutations, and that these lesions have a distinct pattern of genomic evolution en route to malignancy. Further work is needed to identify additional high-risk early genomic events that lead to the development of GE adenocarcinomas, as this may enhance our ability to risk stratify patients with cancer precursor lesions, such as BE. Models of GE adenocarcinomas and CAC will benefit from the incorporation of early *TP53* mutations and relevant environmental exposures to better capture complex features of cancer initiation and progression. The early onset and evolution of CIN in GE cancer enabled by initiating *TP53* mutations promote amplification of unaltered genomic loci associated with MAPK and cell cycle pathways, providing an adaptive fitness advantage. Focal CIN also imparts potential therapeutic vulnerabilities, especially to DDR pathway inhibitors that are currently being tested in preclinical and clinical settings. We hope that through a better understanding of disease mechanisms, we will be able to define the population of patients with gastric and esophageal adenocarcinomas that will benefit from DDR pathway inhibitors.

## Figures and Tables

**Figure 1 cancers-13-05915-f001:**
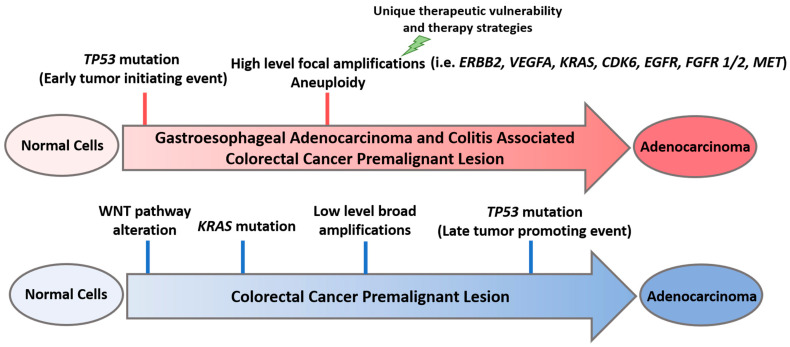
Early *TP53* mutations and high-level focal amplifications in gastroesophageal (GE) adenocarcinoma and colitis-associated colorectal cancer (CAC) premalignant lesion, as opposed to late *TP53* mutations and low-level broad amplifications and mutations in colorectal cancer (CRC) premalignant lesion. *ERBB2:* Erb-B2 receptor tyrosine kinase 2; *VEGFA*: vascular endothelial growth factor A; *KRAS:* Kirsten rat sarcoma; *CDK6*: cyclin dependent kinase 6; *EGFR*: epidermal growth factor receptor; *FGFR1/2*: fibroblast growth factor receptor 1/2; *MET*: mesenchymal epithelial transition factor.

**Figure 2 cancers-13-05915-f002:**
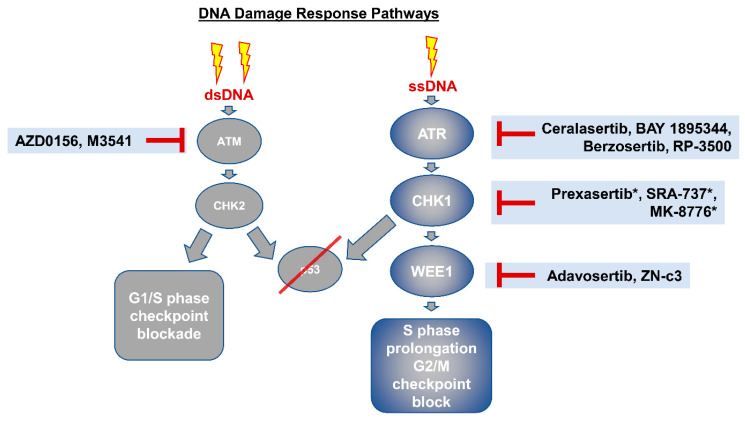
DNA damage response (DDR) and putative inhibitors of potential clinical importance. Single lightning bolt indicates damage to one strand of DNA. Double lightning bolt indicates damage to both strands of DNA. Upon DNA damage, the pathway is activated, leading to either G1/S phase checkpoint blockade or S phase prolongation and prevention of growth/mitosis. Repair of the DNA strand breaks lead to p53 activation, but in those cells with *TP53* alterations, checkpoints are bypassed. Targeting the DDR pathway upstream of p53 will theoretically lead to cell death due to mitotic catastrophe. dsDNA: double stranded DNA; ssDNA: single-stranded DNA; G1/S: restriction checkpoint before DNA synthesis phase in cell cycle; S phase: synthesis phase; G2/M: growth phase checkpoint leading to mitotic phase. * Given the molecular similarities of CHK1 and CHK2, CHK1 inhibitors also have varying degrees of activity against CHK2. dsDNA: double stranded DNA; ssDNA: single-stranded DNA; G1/S: restriction checkpoint before DNA synthesis phase in cell cycle; S phase: synthesis phase; G2/M: growth phase checkpoint leading to mitotic phase.

**Table 1 cancers-13-05915-t001:** Potential therapeutic vulnerabilities associated with chromosomal instability and high-level focal amplifications in gastroesophageal (GE) adenocarcinoma. Food and Drug Administration-approved drugs for gastroesophageal cancer are indicated by *. Aside from trastuzumab deruxtecan and bemarituzumab, most agents listed in this table have, thus far, not shown single-agent efficacy clinically, suggesting that combinatorial approaches may be needed [3,9].

High Level Focal Amplification	Prevalence in GE Adenocarcinoma	Potential Therapeutic Agent	Trial References
*ERBB2*	24–32%	trastuzumab ^3,^*, lapatinib ^3^, neratinib ^1^, tucatinib ^0^, trastuzumab deruxtecan ^2,^*	[22,23,24,25]
*VEGFA*	28%	VEGF inhibitors (ramucirumab ^3,*^, Lenvatinib ^2^)	[26,27,28,29,30,31,32,33,34]
*KRAS*	13–17%	MEK inhibitors (binimetinib ^0^, cobimetinib ^0^), ERK inhibitors ^0^, RAF inhibitors ^0^	-
*CDK6*	14%	palbociclib ^2^, abemaciclib ^0^, ribociclib ^0^	[35]
*EGFR*	10%	cetuximab ^2^, panitumumab ^3^, ABT-806 ^2^	[36,37,38,39]
*FGFR1/2*	8–10%	bemarituzumab ^2^	[40,41]
*MET*	8%	crizotinib ^1^, capmatinib ^1^, tepotinib ^1^	[42,43,44]

^1,2,3^: Phase of clinical trials in gastroesophageal cancer; ^0^: no published data or clinical trials in patients with gastroesophageal cancer. *ERBB2:* Erb-B2 receptor tyrosine kinase 2; *VEGFA*: vascular endothelial growth factor A; *KRAS:* Kirsten rat sarcoma; *CDK6*: cyclin dependent kinase 6; *EGFR*: epidermal growth factor receptor; *FGFR1/2*: fibroblast growth factor receptor 1/2; *MET*: mesenchymal epithelial transition factor.

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
