# Peer review of "Early TP53 Alterations Shape Gastric and Esophageal Cancer Development"

_cancers, 2021, doi:10.3390/cancers13235915_

Round 1

Reviewer 1 Report

In this perspective article, the authors discussed their recent finding regarding the early TP53 mutations that enables the progression of stomach and esophageal adenocarcinomas. In contrast, the early TP53 mutations do not occur in other GI cancers such as colorectal and pancreatic adenocarcinomas. The overall review and perspectives are nice. What lacking is the signaling network that regulates TP53 mutations. Also, outstanding questions should be provided at the end of the article. These will allow readers to have real perspectives on future directions. In particular, the authors should have looked into a better scenario for the upstream protein signaling and alterations that could lead to TP53 gene instability and mutation. According to the literature, tumor suppressor WWOX instability and functional deficiency could lead to alterations of numerous genes including TP53 (J Biol Chem. 2005;280:43100-8; Trends Mol Med. 2007;13:12-22; Cancer Res. 2016;76:6107-6117; Cell Cycle. 2019;18:1177-1186)

Other Concerns:

1) It appears that gastric and esophageal (GE) adenocarcinomas share the same molecular pathway and TP53 mutation in the early state is a critical point in causing GE adenocarcinomas. I’m wondering if there is a mouse model that exhibits GE adenocarcinomas in a synchronic manner.

2) It is good to know that clinical trials are ongoing for GE adenocarcinomas. Does the efficacy show similar results in gastric and esophageal adenocarcinomas if patients have both?

3) In addition to TP53, are there any gene risk factors good for prognosis and diagnosis in the early-stage of GE adenocarcinomas?

4) Page 3, Section 2:  “…The rates of progression from BE to esophageal adenocarcinoma are quite low[12]; the rates of progression in gastric intestinal metaplasia are challenging to measure[13] but are also thought to be infrequent…..”  Confusing. Please rephrase.

5) “Helicobacter Pylori” should be “Helicobacter pylori”.

6) Please make a new graph detailing genes regulated by the DDR pathway, which contributes to early TP53 mutations in both gastric and esophageal cancers, and links to clinical treatments and/or therapeutic interventions.

7). All gene names in the text should be italic. For example, TP53, MET, FGFR2, HER2 and nine others.

Author Response

Reviewer 1
In this perspective article, the authors discussed their recent finding regarding the early TP53 mutations that enables the progression of stomach and esophageal adenocarcinomas. In contrast, the early TP53 mutations do
not occur in other GI cancers such as colorectal and pancreatic adenocarcinomas. The overall review and
perspectives are nice. What lacking is the signaling network that regulates TP53 mutations. Also, outstanding
questions should be provided at the end of the article. These will allow readers to have real perspectives on future
directions. In particular, the authors should have looked into a better scenario for the upstream protein signaling
and alterations that could lead to TP53 gene instability and mutation. According to the literature, tumor suppressor
WWOX instability and functional deficiency could lead to alterations of numerous genes including TP53 (J Biol
Chem. 2005;280:43100-8; Trends Mol Med. 2007;13:12-22; Cancer Res. 2016;76:6107-6117; Cell
Cycle. 2019;18:1177-1186)

A new figure (Figure 2) has been added to the article indicating the classical upstream TP53 genes (DDR genes)
and therapeutic interventions linked with them in GE adenocarcinoma. Important outstanding question added at
the end of article along with relevant information about WWOX in the section 2 as suggested by the reviewer.

Other Concerns:

1) It appears that gastric and esophageal (GE) adenocarcinomas share the same molecular pathway
and TP53 mutation in the early state is a critical point in causing GE adenocarcinomas. I’m wondering if there is a
mouse model that exhibits GE adenocarcinomas in a synchronic manner.

To our knowledge, a mouse model that develops synchronous gastric and esophageal adenocarcinomas has not
been reported. There are a few genetically engineered mouse-models of gastroesophageal junction (GEJ)
cancer.

Park, J. W., Park, D. M., Choi, B. K., Kwon, B. S., Seong, J. K., Green, J. E., Kim, D.-Y. and Kim, H. K.
(2015). Establishment and characterization of metastatic gastric cancer cell lines from murine gastric
adenocarcinoma lacking Smad4, p53, and E-cadherin. Mol. Carcinog. 54, 1521-1527.

Tu, S., Bhagat, G., Cui, G., Takaishi, S., Kurt-Jones, E. A., Rickman, B., Betz, K. S., Penz-Oesterreicher,
M., Bjorkdhl, O., Fox, J. G., et al. (2008). Overexpression of interleukin-1beta induces gastric
inflammation and cancer and mobilizes myeloid-derived suppressor cells in mice. Cancer Cell 14, 408-
419.

2) It is good to know that clinical trials are ongoing for GE adenocarcinomas. Does the efficacy show similar
results in gastric and esophageal adenocarcinomas if patients have both?

Genomically, gastric and esophageal adenocarcinomas are very similar. Adenocarcinoma can be identified
anywhere along the esophagus into the stomach. Sometimes, it can present at the gastroesophageal junction or
involve the esophagus and stomach. In metastatic disease, clinical oncologists treat gastric, esophageal and
GEJ in the same way, and the treatment is similar across site of origin. We have added new text regarding this in
section 5.

3) In addition to TP53, are there any gene risk factors good for prognosis and diagnosis in the early-stage of GE
adenocarcinomas?

There are no widely accepted genes or genomic features that have been identified as prognostic comparing early-
stage to late-stage GE adenocarcinomas. Some genes such as GATA6, MUC6, and TP53 have been implicated
as prognostic overall, but they are not currently part of the diagnostic schema (
https://pubmed-ncbi-nlm-nih-
gov.treadwell.idm.oclc.org/34206291/
). We added a sentence at the end of the article stating that more work is
needed to identify these high-risk genomic features. This also addresses the reviewer suggestion to indicate
outstanding questions in the field.

4) Page 3, Section 2: “...The rates of progression from BE to esophageal adenocarcinoma are quite low[12]; the
rates of progression in gastric intestinal metaplasia are challenging to measure[13] but are also thought to be
infrequent.....” Confusing. Please rephrase.

Thanks for this suggestion. We adjusted the language to clarify.

5) “Helicobacter Pylori” should be “Helicobacter pylori”.
Thanks for pointing this out. We made this change.

6) Please make a new graph detailing genes regulated by the DDR pathway, which contributes to early TP53
mutations in both gastric and esophageal cancers, and links to clinical treatments and/or therapeutic
interventions.

A new figure (Figure 2) indicating the above points has been added to the article.

7). All gene names in the text should be italic. For example, TP53, MET, FGFR2, HER2 and nine others.

We apologize for this oversight. We have made these corrections.

Reviewer 2 Report

The authors claim that TP53 mutations occur in the early phase of GE adenocarcinoma development in compared with colorectal cancers and this early TP53 alteration defines the unique cancer properties. The manuscript is well written and also gives a somewhat fresh inspiration. Still I have several concerns as follows;

  1. The authors explains TP53 mutations limited in certain type (CIN type) of gastric cancer. However, TP53 mutations are present in other types of gastric cancers including EBV, MSI, and genomically stable. Is TP53 mutation an early event as well in these other molecular types? In this regard, do you think your hypothesis can be generalized or just be limited in certain type of GCs? Authors should clarify these throughout the manuscript including title.

  1. In the paragraph of “4. Early TP53 mutations shape the method~”, the association of focal amplification and early TP53 mutation was already mentioned in the earlier paragraph. We see KRAS-mut CRCs are also resistant to MAPK inhibitors. I think your description may give a wrong idea to the readers. Collectively, I don’t think this whole paragraph is necessary.

  1. I wish the authors add more details in the Figure 1. For example, associated molecular type, gene names for focal amplifications, specific genes related to the therapeutic vulnerability, and the frequencies of specific genetic alterations need to be addressed.    

Author Response

Reviewer 2
The authors claim that TP53 mutations occur in the early phase of GE adenocarcinoma development in compared
with colorectal cancers and this early TP53 alteration defines the unique cancer properties. The manuscript is well
written and also gives a somewhat fresh inspiration. Still I have several concerns as follows;

1) The authors explains TP53 mutations limited in certain type (CIN type) of gastric cancer. However, TP53
mutations are present in other types of gastric cancers including EBV, MSI, and genomically stable. Is TP53
mutation an early event as well in these other molecular types? In this regard, do you think your hypothesis
can be generalized or just be limited in certain type of GCs? Authors should clarify these throughout the
manuscript including title.

When compared to
the frequency in CIN (71%), TP53 alterations are substantially less frequent in other gastric
cancer subtyp
es: EBV 7%, MSI 33%, GS 12% 1. Even though it is mutated in these subtypes, it is
unclear i
f the mutation is an early event like it is in CIN. For these reasons, the perspective is largely referring
to TP53 mutations in th
e CIN subtype of gastric and esophageal adenocarcinomas. Here is the sentence from
the first paragraph that best summarizes this:
While a comprehensive molecular analysis of gastrointestinal
adenocarcinomas revealed five molecular subtypes tha
t largely transcended anatomic boundaries1, the
similarity in molecular composition of GE adenocarcinomas is largely due to the high proportion of the

chromosomal instability
(CIN) subtype, which accounts more than half of gastric adenocarcinomas and almost
90% of esophageal adenocarcinomas
.

2) In the paragraph of “4. Early TP53 mutations shape the method~”, the association of focal amplification and
early TP53 mutation was already mentioned in the earlier paragraph. We see KRAS-mut CRCs are also
resistant to MAPK inhibitors. I think your description may give a wrong idea to the readers. Collectively, I don’t
think this whole paragraph is necessary.

Thank you for raising this concern
. We do agree that KRAS-mut CRC response to MAPK inhibition is far from
optimal. That being said
, there is preclinical evidence that WT KRAS amplification and KRAS-mut CRC respond
differen
tly to MAPK inhibition, which is cited in this paragraph. Furthermore, we think it is important to mention
some of the molecular
mechanism that distinguish these alterations. As newer MAPK treatment

3) I wish the authors add more details in the Figure 1. For example, associated molecular type, gene names for
focal amplifications, specific genes related to the therapeutic vulnerability, and the frequencies of specific
genetic alterations need to be addressed.

We added
more detail to Figure 1 and added a new table (Table 1) to include the names of genes with high
level focal amplif
ication, their prevalence in GE adenocarcinoma and corresponding potential therapeutic
agents
including FDA approved ones.

1. Liu Y, Sethi NS, Hinoue T, et al. Comparative Molecular Analysis of Gastrointestinal Adenocarcinomas.
Cancer Cell 2018;33:721-735 e8.

Round 2

Reviewer 1 Report

Accepted.

Author Response

Thank you